# Characterization of *Escherichia coli* Cefotaxime-Resistance in Al-Ahsa, KSA: Predominance of CTX-15 and First Report of $bla_{CMY\text{-}42}$ Gene

Melek Ben Aissa [1,*], Sana Ferjani [2], Mohamed Salah Abassi [3], Nada Al-Suwailem [4] and Ilhem Boutiba [2,5]

1. Applied College in Abqaiq, King Faisal University, Al-Ahsa 31982, Saudi Arabia
2. Laboratory of "Resistance to Antimicrobiens" LR99ES09, College of Medicine, Tunis El Manar University, Tunis 1007, Tunisia
3. Laboratory of Bacteriological Research, Tunisian Institute of Veterinary Research, Tunis El Manar University, Tunis 1006, Tunisia
4. Chemistry Department, College of Science, King Faisal University, Al-Ahsa 31982, Saudi Arabia
5. Laboratory of Microbiology, Charles-Nicolle Hospital, Tunis 1006, Tunisia
* Correspondence: maissa@kfu.edu.sa; Tel.: +966-135891030

**Abstract:** We determined an antibiotic resistance mechanism in the eastern region, KSA, and the genetic factor clonal relatedness within Gram-negative bacteria. During our retrospective study, a total number of 29 *E. coli* ESBL producer strains were isolated for patients visiting King Fahad Hospital, Al-Ahsa, KSA. The *bla* genes were detected via PCR and identified via sequencing. Associated plasmid-mediated quinolone resistance genes, as well as *int1* and *int2* genes, were also studied. Phylogenetic groups, the ST131 clone, virulence factors, and PFGE were also checked. The $bla_{CTX\text{-}M\text{-}9}$ (3.7%), $bla_{CTX\text{-}M\text{-}27}$ (22.2%), and $bla_{CTX\text{-}M\text{-}15}$ (77.8%) genes were identified; however, the $bla_{CMY\text{-}42}$ (7.4%) gene was recorded for the first time in KSA. The *qnrS1* gene was found in 44.4% of strains, and among them, 50% concomitantly harbored the *aac(6′)Ib-cr*. The *int1* gene was detected in 25.9% strains; nonetheless, the *int2* gene was found in 7.4% of isolates. The strains belonged mainly to the B2 and D phylogroups. PFGE showed unrelated patterns. Some isolates belonged to the pandemic clone ST131. We describe a large dissemination of antibiotic resistance to third-generation cephalosporins in the eastern region, KSA, with the occurrence of the $bla_{CMY\text{-}42}$ gene. The clone ST131 seems to be the principal contributor for $bla_{CTX\text{-}M\text{-}15}$ gene spread.

**Keywords:** antibiotic resistance; ESBL; *E. coli*; CTX-M-15; CMY-42; ST131; KSA

## 1. Introduction

Secondary infections and organ failure are the graver consequences observed worldwide during COVID-19, causing a high mortality rate [1]. Secondary infections specifically associated with SARS-CoV-2 are still poorly known; however, many studies have shown that co-infection factors include *Salmonella*, *Shigella*, and *E. coli* [2–4]. On the other hand, human extra-intestinal diseases, including urinary tract infection (UTI), bacteremia, and meningitis can be caused by a group of pathogens such as *Escherichia coli* [5]. *E. coli*, a producer of ESBLs, is recorded among the six highest multi-resistant germs that are particularly curable with only a few effective drugs [6]. It is frequently correlated with high healthcare costs, morbidities, and mortalities [7,8]. In 2017, the estimated number of critical cases of infection with pathogens producing ESBLs was about 200,000 with 9100 deaths; however, the total healthcare cost was estimated to be USD 1.2B [9]. Furthermore, numerous investigations have confirmed that the ESBL, exhibited among *E. coli*, is the major mechanism reported for the β-lactam resistance [10,11]. Over the last two decades, the fast diffusion of ESBLs has emerged globally, with the development of a worldwide outbreak being a powerful menace to human health, while continuous surveys and molecular characterizations of these strains have been conducted in healthcare units.

The discovery of pharmaceutical compounds with new antibacterial properties used to fight these bacterial infections seems to have been dropped, due to the vicious circle of new resistance for each new antibiotic discovered [12]. On the other hand, the high number of intra-travelers and extra-travelers in KSA has contributed to the wide dissemination of antibiotic resistance, making antibiotic prescriptions a complicated process, especially when it is due to bacterial co-infections or complications in healthcare units for critical clinical cases, particularly throughout the COVID-19 pandemic [13].

The spread of ESBLs is related to the diversity of enzymes and transferable genetic elements, and the capacity for a large spread of clones [14]. Indeed, a significantly large amount of published data in different countries [15] has indicated that the ST131 clone producing CTX-15 was associated with noticeable epidemiological changes in hospitals and communities for infectious diseases [16]. In KSA, a few epidemiological data on pathogens producing ESBLs were given, and seemed to be mainly focused on Riyadh and Dammam.

In this observational study, we determined the implication of ESBL genetic factors and plasmids. We assessed the clonal relatedness and virulence factors among a collection of *E. coli* samples collected between November 2016 and March 2017 in King Fahad Hospital (KFH), Al-Ahsa in KSA.

## 2. Materials and Methods

### 2.1. Study Design

The research was observational, cross-sectional, and retrospective. Convenience sampling for 5 months during the winter season between November 2016 and March 2017 was utilized (Figure 1). The sample size was calculated using Statdirect software version 26 (CI 95%). For inclusion criteria, only *E. coli* strains and positive ESBLs were maintained in the study. However, the exclusion criteria were the high sensitivity to cephalosporin of third-generation (C3G), non-ESBL producers, and not an *E. coli* strain. EC10 and EC13 were excluded for a loss of stable C3G resistance characteristics after a few generations.

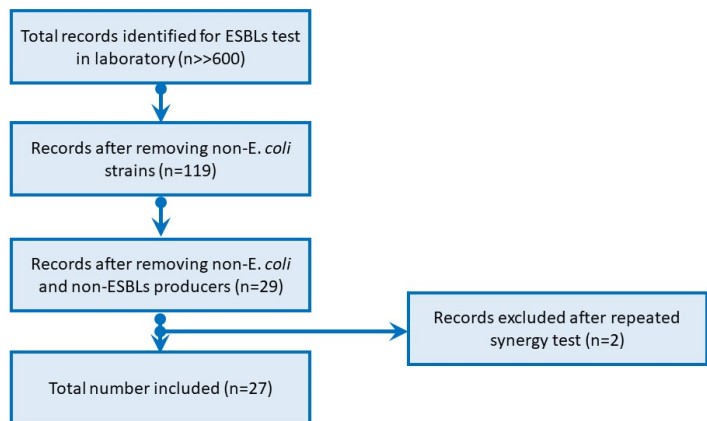

**Figure 1.** Study design flow diagram (records were approved after repeated antibiograms and synergy tests. Results were compared to our inclusion criteria.).

### 2.2. Antimicrobial Susceptibility Tests and Cefotaxime Resistance Transfer Assays

Antimicrobial susceptibilities were determined by conducting the disk diffusion technique according to the European Committee on Antimicrobial Susceptibility Testing guidance (CA-SFM/EUCAST, 2016). A total of 22 antibiotics (Bio-Rad, Marnes-la-Coquette, France) were used according to the CA-SFM guidelines (amoxicillin/clavulanic acid, ceftazidime, cefotaxime, cefoxitin, cefepime, ertapenem, nalidixic acid, ciprofloxacin, sulfamethoxazole/trimethoprim, tetracycline, minocycline, gentamicin, tobramycin, netilmicin, amikacin, norfloxacin, tigecycline, colistin, nitrofurantoin, fosfomycin, chloramphenicol, and aztreonam). The minimum inhibitory concentration was determined for cefotaxime and ceftazidime using the agar diffusion method, following the Clinical and Labora-

tory Standards Institute. ESBLs were distinguished using the test of synergy (TS) within amoxicillin/clavulanic acid and ceftazidime, cefotaxime, aztreonam, or cefepime (with and without cloxacillin at 250 mg/L). The control strains used were ATCC700603 and ATCC25922 (EUCAST, 2015). Conjugation experiments utilizing *E. coli* J53 (Rifampin resistant) as a receiver were performed to assess the transferability of C3G resistance. Our isolates were initially selected on Mueller Hinton (MH) agar (Oxoid Ltd., Basingstoke, UK) plates incorporating rifampicin within 400 mg/L. Resistance transfer experiments were implemented on brain–heart infusion broth (BHI) (Oxoid Ltd., Basingstoke, UK) for 18 h at 37 °C, and the obtained transconjugants were grown on MH agar plates after adding both cefotaxime (2 mg/L) and rifampicin (400 mg/L). The transconjugants' antimicrobial susceptibilities were tested using a previously mentioned method. The strain origins were recorded (Table 1). Frequencies were calculated using SPSS software version 26.

### 2.3. Characterization of Antibiotic Resistance Genes

The strains exhibiting positive TS were evaluated via PCR for the occurrence of $bla_{TEM}$, $bla_{SHV}$, $bla_{CTX-M}$, $bla_{PER}$, $bla_{GES}$, and $bla_{VEB}$ genes (Table 2). Furthermore, the screenings for cephalosporinases were aimed at $bla_{FOX}$, $bla_{ACC}$, $bla_{CIT}$, $bla_{DHA}$, $bla_{EBC}$, and $bla_{MOX}$ genes, as described previously (Dallenne et al., 2010). Amplified DNA fragments were sequenced, applying a DNA sequencer (ABI PRISM 3130, Applied Biosystems, Foster City, CA, USA) [17], and then compared in the GenBank database.

### 2.4. Quinolone Resistance Genes, Integrons, and Plasmid Incompatibility Groups

Quinolone resistance genes (QR) were performed via multiplex PCR assays [18]. The screened genes are listed in Table 3, as well as integrase genes [19]. Furthermore, plasmid incompatibilities groups (Inc.) were established (Table 4) as described [20].

### 2.5. Phylogenetic Analysis, Genetic Relatedness, and ST131 Identification

Triplex PCRs for the phylogenetic groups A, B1, B2, and D, respectively, were screened [20]. Furthermore, virulence genotypes encoding toxins, invasins, adhesins, and siderophores were assessed, applying multiplex PCR (Table 3). The control isolates CFT073 and J96 were used [5,21,22]. The *XbaI* restriction enzyme was used for Pulsed-Field Gel Electrophoresis (PFGE), it was performed to assess the clonal correlation of different isolates, as reported previously [23]. ST131 was researched using an O25b-specific PCR method, with *pabB* and *trpA* allele-specific primers for the B2 phylogenetic group strains [24].

**Table 1.** Patients' demographic data and specimen types included in our study.

|  | Variables | Numbers | Percentage (%) |
|---|---|---|---|
| Gender | Male | 13 | 48.1 |
|  | Female | 14 | 51.9 |
| Age | ≤37 | 5 | 18.5 |
|  | 38–55 | 12 | 44.4 |
|  | 56–73 | 6 | 22.2 |
|  | 74+ | 4 | 14.8 |
| Specimen | Sputum | 2 | 7.4 |
|  | Surgical tissue | 2 | 7.4 |
|  | Urine | 14 | 51.9 |
|  | Wound | 9 | 33.3 |

### 2.6. Ethical Approval

The deanship of scientific research at King Faisal University, Al-Ahsa, KSA approved the study (ref. no. EA000528). The information and details collected were confidential. No personal information was shared. The study was a secondary analysis to a routine laboratory test.

## 3. Results

### 3.1. Antibiotic Susceptibilities and Cefotaxime Resistance Transfer Assays

During the winter season, the patterns of *E. coli* ESBL producers were 51.9% female, the mean age was 54.78, with 44.4% aged between 39 and 55 years; and urine was the prevalent specimen type, at 51.9% (Table 1). The multidrug resistance characterization indicated the production of both ESBLs and non-ESBLs among our collection (Table 2). A high resistance rate for the tested antibiotics was observed for amoxicillin/clavulanic acid, cefotaxime, and ceftazidime (Table 2). However, all the strains were susceptible to amikacin, colistin, and carbapenems. The isolates were resistant to gentamicin (18.5%; 5/27), tobramycin (33.3%; 9/27), netilmicin (7.4%; 2/27), nalidixic acid (100%; 27/27), ciprofloxacin (88.8%; 24/27), tetracycline (55.5%; 15/27), minocycline (14.8%; 4/27), fosfomycin (100%; 27/27), chloramphenicol (70.3%; 19/27), and trimethoprim/sulfamethoxazole (85.2%; 23/27) (Table 2). ESBL phenotypes were found in 26 isolates, and plasmidic cephalosporinases (pAmpC) were identified in two strains. However, one strain co-produced both an ESBL and a cephalosporinase. A successful transfer of cefotaxime resistance was detected for only 12 isolates (Table 5).

**Table 2.** *E. coli* strains characteristics and associated antibiotic resistances.

| Strain | β-Lactam Resistance Phenotypes Detected * | Non-β-Lactam Resistance Phenotypes Detected * |
|---|---|---|
| EC 1 | CAZ, CTX, FEP | NAL, CIP, TET, FOS, SXT |
| EC 2 | AMC, CAZ, CTX, FEP | TOB, NAL, CIP, FOS, CHL, SXT |
| EC 3 | AMC, CTX, FEP | NAL, TET, FOS, CHL, SXT |
| EC 4 | AMC, CTX | NAL, CIP, TET, FOS, CHL, SXT |
| EC 5 | AMC, CTX, FEP | NAL, TET, MNO, FOS, CHL, SXT |
| EC 6 | AMC, CAZ, CTX, FEP | NAL, CIP, TET, MNO, FOS, CHL, SXT |
| EC 7 | AMC, CTX | NAL, CIP, FOS |
| EC 8 | AMC, CAZ, CTX, FOX | TOB, NAL, CIP, TET, MNO, FOS, CHL, SXT |
| EC 9 | AMC, CAZ, CTX, FEP | NAL, CIP, FOS, CHL, SXT |
| EC 11 | AMC, CAZ, CTX, FEP | GMN, TOB, NAL, CIP, TET, FOS, SXT |
| EC 12 | AMC, CAZ, CTX, FEP | NAL, FOS, SXT |
| EC 14 | AMC, CAZ, CTX, FEP | GMN, TOB, NET, NAL, CIP, TET, FOS, CHL, SXT |
| EC 15 | AMC, CTX, FEP | NAL, CIP, FOS, CHL, SXT |
| EC 16 | AMC, CAZ, CTX, PEP | GMN, TOB, NAL, CIP, TET, FOS, CHL, SXT |
| EC 17 | AMC, CTX, FEP | NAL, CIP, TET, FOS, CHL, SXT, |
| EC 18 | CAZ, CTX, FEP | NET, NAL, CIP, FOS, CHL, |
| EC 19 | AMC, CTX, FEP | NAL, CIP, FOS |
| EC 20 | CTX, FEP | NAL, CIP, FOS, SXT |
| EC 21 | AMC, CAZ, CTX, FEP | NAL, CIP, TET, MNO, FOS, SXT |
| EC 22 | AMC, CTX | NAL, CIP, TET, FOS, CHL, SXT |
| EC 23 | AMC, CTX | NAL, CIP, FOS, CHL, SXT |
| EC 24 | AMC, CTX | TOB, NAL, CIP, FOS, CHL, SXT |
| EC 25 | AMC, CTX, FEP | TOB, NAL, CIP, FOS, SXT |
| EC 26 | AMC, CTX, FEP | NAL, CIP, TET, FOS, CHL |
| EC 27 | AMC, CAZ, CTX, FEP | GMN, TOB, NAL, CIP, TET, FOS, CHL, SXT |
| EC 28 | AMC, CAZ, CTX, FEP | GMN, TOB, NAL, CIP, TET, FOS, CHL, SXT |
| EC 29 | AMC, CTX, FOX | NAL, CIP, FOS, CHL, SXT |

\* AMC: amoxicillin/clavulavic acid (30 µg); CAZ: ceftazidime (10 µg); CTX: cefotaxime (30 µg); FOX: cefoxitine (30 µg); FEP: cefepime (30 µg); NAL: nalidixic acid (30 µg); CIP: ciprofloxacin (5 µg); SXT: sulfamethoxazole/trimethoprim (25 µg); TET: tetracycline (30 µg); MNO: minocycline (30 µg); GMN: gentamicin (10 µg); TOB: tobramycin (10 µg); NET: netilmicin (10 µg); FOS: fosfomycin (50 µg); CHL: chloramphenicol (30 µg).

### 3.2. Characterization of Antibiotic Resistance

The CTX-15 enzyme was found in 77.8% (21/27) strains; however, CTX-27 was recorded in 22.2% (6/27), and CTX-9 β-lactamase was less frequently presented, with only 3.7% (1/27) of the strains. The $bla_{TEM-1}$ gene was identified in 74.1% (20/27) strains, while the $bla_{SHV-1}$ gene was detected in 18.5% (5/27) strains (Table 1). The gene sets $bla_{CMY-42}/bla_{CTX-M-15}/bla_{SHV-1}$ and $bla_{CMY-42}/bla_{TEM-1}/bla_{SHV-1}$ were identified as co-ESBL

producers. The $bla_{CTX\text{-}M\text{-}15}$/$bla_{TEM\text{-}1}$ gene association was observed for 48.14% (13/27) of the strains, while the $bla_{CTX\text{-}M\text{-}27}$/$bla_{TEM\text{-}1}$ was detected in 11.1% (3/27). Both the $bla_{CTX\text{-}M\text{-}15}$/$bla_{SHV\text{-}1}$ genes were seen in 7.4% (2/27) of the strains. The association of $bla_{CMY\text{-}42}$/ $bla_{CTX\text{-}M\text{-}15}$/$bla_{SHV\text{-}1}$, $bla_{CTX\text{-}M\text{-}15}$/$bla_{CTX\text{-}M\text{-}27}$/$bla_{TEM\text{-}1}$, $bla_{CTX\text{-}M\text{-}9}$/$bla_{CTX\text{-}M\text{-}27}$/ $bla_{TEM\text{-}1}$, $bla_{CTX\text{-}M\text{-}15}$/$bla_{TEM\text{-}1}$/$bla_{SHV\text{-}1}$, and $bla_{CMY\text{-}42}$/$bla_{TEM\text{-}1}$/$bla_{SHV\text{-}1}$ genes was reported in 3.7% for each set (1/27) (Table 3).

**Table 3.** Molecular characterization, virulence profiles, and virulence scores of *E. coli* strains.

| Strains | *bla* and PMQR* Genes Identified | Virulence Profile | Virulence Score |
|---|---|---|---|
| EC 1 | $bla_{CTX\text{-}M\text{-}15}$, *qnrS1* | *fimH-papGII-iha-iutA-traT-malX-usp-ompT* | 8 |
| EC 2 | $bla_{CMY\text{-}42}$, $bla_{CTX\text{-}M\text{-}15}$, $bla_{SHV\text{-}1}$ | *fimH-papGII-iutA-malX* | 4 |
| EC 3 | $bla_{CTX\text{-}M\text{-}15}$, $bla_{CTX\text{-}M\text{-}27}$, $bla_{TEM}$ | *papGII-ompT* | 2 |
| EC 4 | $bla_{CTX\text{-}M\text{-}27}$ | *fimH-papGII-iha-malX-usp-ompT* | 6 |
| EC 5 | $bla_{CTX\text{-}M\text{-}15}$, $bla_{TEM\text{-}1}$ | *malX-ompT* | 2 |
| EC 6 | $bla_{CTX\text{-}M\text{-}15}$, $bla_{TEM\text{-}1}$ | *papGII-fyuA-malX-ompT* | 4 |
| EC 7 | $bla_{CTX\text{-}M\text{-}15}$, *qnrS1* | *fimH-papGII-iha-iutA-kpsMTII-malX-usp-ompT* | 8 |
| EC 8 | $bla_{CMY\text{-}42}$, $bla_{TEM\text{-}1}$, $bla_{SHV\text{-}1}$, *qnrS1*, *aac(6′)Ib-cr* | *fimH-papGII* | 2 |
| EC 9 | $bla_{CTX\text{-}M\text{-}15}$, $bla_{TEM\text{-}1}$ | *papGII-malX* | 2 |
| EC 11 | $bla_{CTX\text{-}M\text{-}15}$, $bla_{SHV}$, *qnrS1* | *papGII-iha-traT-malX-usp-ompT* | 6 |
| EC 12 | $bla_{CTX\text{-}M\text{-}15}$, *qnrS1*, *aac(6′)Ib-cr* | *papGI-iha-traT-malX-usp* | 5 |
| EC 14 | $bla_{CTX\text{-}M\text{-}15}$, $bla_{TEM\text{-}1}$ | *papGIII-malX-usp-ompT* | 4 |
| EC 15 | $bla_{CTX\text{-}M\text{-}15}$ | *papGII-iha-usp-ompT* | 4 |
| EC 16 | $bla_{CTX\text{-}M\text{-}15}$, $bla_{TEM\text{-}1}$, $bla_{SHV\text{-}1}$ | *papGII-iha-iutA-traT-usp-ompT* | 6 |
| EC 17 | $bla_{CTX\text{-}M\text{-}15}$, $bla_{TEM\text{-}1}$ | *fimH-papGII-iha–iutA-ompT* | 5 |
| EC 18 | $bla_{CTX\text{-}M\text{-}15}$, $bla_{TEM\text{-}1}$ | *fimH-iha-iutA-kpsMTII-malX-usp-ompT* | 7 |
| EC 19 | $bla_{CTX\text{-}M\text{-}15}$, $bla_{TEM\text{-}1}$, *qnrS1* | *fimH-iha-iutA-kpsMTII-malX-usp-ompT* | 7 |
| EC 20 | $bla_{CTX\text{-}M\text{-}27}$, $bla_{TEM\text{-}1}$, *qnrS1* | *fimH-papGII-iha-traT-usp-ompT* | 6 |
| EC 21 | $bla_{CTX\text{-}M\text{-}15}$, $bla_{TEM\text{-}1}$, *qnrS1* | *fimH-iha-kpsMTII-usp-ompT* | 5 |
| EC 22 | $bla_{CTX\text{-}M\text{-}27}$, $bla_{TEM\text{-}1}$ | *fimH-papGII-iha-usp-ompT* | 5 |
| EC 23 | $bla_{CTX\text{-}M\text{-}27}$, $bla_{TEM\text{-}1}$ | *papGII-ompT* | 2 |
| EC 24 | $bla_{CTX\text{-}M\text{-}15}$, $bla_{TEM\text{-}1}$, *qnrS1*, *aac(6′)Ib-cr* | - | - |
| EC 25 | $bla_{CTX\text{-}M\text{-}15}$, $bla_{SHV\text{-}1}$ | *fimH-papGII-iha-iutA-malX-usp-ompT* | 7 |
| EC 26 | $bla_{CTX\text{-}M\text{-}15}$, $bla_{TEM\text{-}1}$ | *fimH-papGII-iutA-ompT* | 4 |
| EC 27 | $bla_{CTX\text{-}M\text{-}15}$, $bla_{TEM\text{-}1}$, *qnrS1*, *aac(6′)Ib-cr* | *papGI-papGII-iha-kpsMTII-hlyA traT-usp-ompT* | 8 |
| EC 28 | $bla_{CTX\text{-}M\text{-}15}$, $bla_{TEM\text{-}1}$-1, *qnrS1*, *aac(6′)Ib-cr* | *fimH-papGI-iha-iutA-kpsMTII-traT-malX-usp-ompT* | 9 |
| EC 29 | $bla_{CTX\text{-}M\text{-}9}$, $bla_{CTX\text{-}M\text{-}27}$, $bla_{TEM\text{-}1}$, *qnrS1*, *aac(6′)Ib-cr* | *fimH-papGII-malX-usp* | 4 |

* PMQR: plasmid-mediated quinolone resistance.

*3.3. Quinolone Resistance Genes, Integrons, and Plasmid Incompatibility*

The *qnrS1* gene was found in 44.44% (12/27) of the strains. The *aac(6′)Ib-cr* variant was observed in 22.22% (6/27) of the strains. The genes *int1* and *int2* were reported, respectively, in 25.9% (7/27) and 7.4% (2/27) for our isolates (Table 3). A total of 48.14% (13/27) of the strains showed a plasmid replicon type. Only *IncF* (11.1%; 3/27) and *IncFIA* (18.5%; 5/27) were identified among our collection; moreover, both co-occurred in 18.5% of isolates (5/27) (Table 3). The transfer of the $bla_{CTX\text{-}M\text{-}15}$/$bla_{TEM\text{-}1}$/*qnrS1*/*aac(6′)Ib-cr* and *int1* genes has been frequently observed (Table 4). Similarly, the *IncF* and *IncFIA* plasmids were also successfully transmitted through conjugation (Table 5).

**Table 4.** Identification of associated plasmid-mediated *int1* and *int2* genes, phylogenetic groups, ST131 clone, and virulence factors and pulsotypes.

| Strains | *int* Genes | PRT | PG | PFGE |
|---------|-------------|-----|-----|------|
| EC 1 | - | F | B2 | P6 |
| EC 2 | *int1* | F | D | P2 |
| EC 3 | *int2* | FIA | D | P20 |
| EC 4 | - | - | B2 | P11 |
| EC 5 | - | - | A | P7 |
| EC 6 | - | - | B2 | P21 |
| EC 7 | - | - | D | P9 |
| EC 8 | *int2* | FIA-F | B1 | P10 |
| EC 9 | *int1* | - | B2 | P8 |
| EC 11 | - | F | B2 | P14 |
| EC 12 | *int1* | FIA-F | B2 | P19 |
| EC 14 | - | - | B2 | P8 |
| EC 15 | *int1* | FIA | B2 | P18 |
| EC 16 | *int1* | - | B2 | P19 |
| EC 17 | - | FIA | B2 * | P1 |
| EC 18 | - | FIA | B2 * | P1 |
| EC 19 | - | FIA | B2 * | P1 |
| EC 20 | - | - | B2 | P16 |
| EC 21 | - | - | B2 | P15 |
| EC 22 | - | - | B2 | P16 |
| EC 23 | - | - | D | P12 |
| EC 24 | - | FIA-F | D | P5 |
| EC 25 | - | - | B2 | P4 |
| EC 26 | - | - | A | P13 |
| EC 27 | *int1* | FIA-F | B2 | P3 |
| EC 28 | *int1* | FIA-F | B2 | P3 |
| EC 29 | - | - | B2 | P1 |

-: Negative character; *: strain belongs to ST131 clone; P: pulsotype; PG: Phylogenetic Group; PRT: Plasmid Replicon Type; PFGE: Pulsed-Field Gel Electrophoresis.

### 3.4. Genetic Relationship, Phylogenetic Groups, and Identification of the ST131 Clone

The B2 phylogenetic group was identified in 70.4% (19/27); it was considered as the main group in our study. Further, the phylogroups D, A, and B1 were presented, respectively, in 18.5% (5/27), 7.4% (2/27), and 3.7% (1/27) of isolates. Three strains amongst the B2 phylogenetic group were positive for the ST131 clone (Table 4). PFGE (Pulsed-Field Gel Electrophoresis) analysis revealed 21 different DNA profiles (P1 to P21) among the 27 isolates. Each pulsotype, P8, P16, and P19 included two isolates, and the pulsotype P1 contained three B2-ST131 isolates. The remainder of the isolates were unrelated (Table 4, Figure 2).

### 3.5. Occurrence of Virulence Genes

Virulence genes were distributed as follows: *fimH* (55.5%, 15/27), *papGI* (11.1%, 3/27), *papGII* (74.1%, 20/27), *fyuA* (3.7%, 1/27), *iha* (59.3%, 16/27), *iutA* (37.1%, 10/27), *ompT* (77.7%, 21/27), *traT* (25.9%, 7/27), *kpsMTII* (22.2%, 6/27), *malX* (55.5%, 15/27), and *usp* (62.9%, 17/27). The virulence scores varied from 0 to 9 (median: 5).

**Table 5.** Positive transconjugants compared with donor strains.

| Strains | Non-β-Lactams Resistance Phenotypes Detected | *bla* and PMQR Genes | *int* Genes | PRT |
|---|---|---|---|---|
| EC 1 | NAL, CIP, TET, FOS, SXT | $bla_{CTX-M-15}$, $bla_{TEM}$, $qnrS1$ | - | F |
| * Tc EC 1 | TET | $bla_{CTX-M-15}$ | - | F |
| EC 7 | NAL, CIP, FOS | $bla_{CTX-M-15}$, $qnrS1$ | - | - |
| Tc EC 7 | - | $bla_{CTX-M-15}$ | - | - |
| EC 8 | TOB, NAL, CIP, TET, MNO, FOS, CHL, SXT | $bla_{CMY-42}$, $bla_{TEM}$, $bla_{SHV}$, $qnrS1$, $aac(6')Ib-cr$ | *int2* | FIA-F |
| Tc EC 8 | TET, MNO | $bla_{CMY-42}$, $bla_{TEM}$, $qnrS1$ | - | FIA |
| EC 11 | GMN, TOB, NAL, CIP, TET, FOS, SXT | $bla_{CTX-M-15}$, $bla_{SHV}$, $qnrS1$ | - | F |
| Tc EC 11 | TET | $bla_{CTX-M-15}$ | - | F |
| EC 12 | NAL, FOS, SXT | $bla_{CTX-M-15}$, $qnrS1$, $aac(6')Ib-cr$ | *int1* | FIA-F |
| Tc EC 12 | - | $bla_{CTX-M-15}$ | *int1* | FIA |
| EC 19 | NAL, CIP, FOS | $bla_{CTX-M-15}$, $bla_{TEM}$, $qnrS1$ | - | FIA |
| Tc EC 19 | - | $bla_{CTX-M-15}$ | - | FIA |
| EC 20 | NAL, CIP, FOS, SXT | $bla_{CTX-M-27}$, $bla_{TEM}$, $qnrS1$ | - | - |
| Tc EC 20 | SXT | $bla_{CTX-M-27}$, $bla_{TEM}$, $qnrS1$ | - | - |
| EC 21 | NAL, CIP, TET, MNO, FOS, SXT | $bla_{CTX-M-15}$, $bla_{TEM}$, $qnrS1$ | - | - |
| Tc EC 21 | TET, MNO, SXT | $bla_{CTX-M-15}$, $bla_{TEM}$, $qnrS1$ | - | - |
| EC 24 | TOB, NAL, CIP, FOS, CHL, SXT | $bla_{CTX-M-15}$, $bla_{TEM}$, $qnrS1$, $aac(6')Ib-cr$ | - | FIA-F |
| Tc EC 24 | TOB, CHL, SXT | $bla_{CTX-M-15}$, $bla_{TEM}$, $qnrS1$, $aac(6')Ib-cr$ | - | FIA-F |
| EC 27 | GMN, TOB, NAL, CIP, TET, FOS, CHL, SXT | $bla_{CTX-M-15}$, $bla_{TEM}$, $qnrS1$, $aac(6')Ib-cr$ | *int1* | FIA-F |
| Tc EC 27 | TET | $bla_{CTX-M-15}$ | *int1* | F |
| EC 28 | GMN, TOB, NAL, CIP, TET, FOS, CHL, SXT | $bla_{CTX-M-15}$, $bla_{TEM}$, $qnrS1$, $aac(6')Ib-cr$ | *int1* | FIA-F |
| Tc EC 28 | TET, SXT | $bla_{CTX-M-15}$, $bla_{TEM}$, $qnrS1$ | *int1* | FIA-F |
| EC 29 | NAL, CIP, FOS, CHL, SXT | $bla_{CTX-M-9}$, $bla_{CTX-M-27}$, $bla_{TEM}$, $qnrS1$, $aac(6')Ib-cr$ | - | - |
| Tc EC 29 | CHL, SXT | $bla_{CTX-M-9}$, $bla_{TEM}$, $qnrS1$ | - | - |

* Tc: transconjugant; -: negative character; NAL: nalidixic acid; CIP: ciprofloxacin; SXT: trimethoprim-sulfamethoxazole; TET: tetracycline; MNO: minocyclin; GMN: gentamicin; TOB: tobramycin; FOS: fosfomicin; CHL: chloramphenicol; PMQR: plasmid-mediated quinolone resistance; PRT: plasmid replicon type.

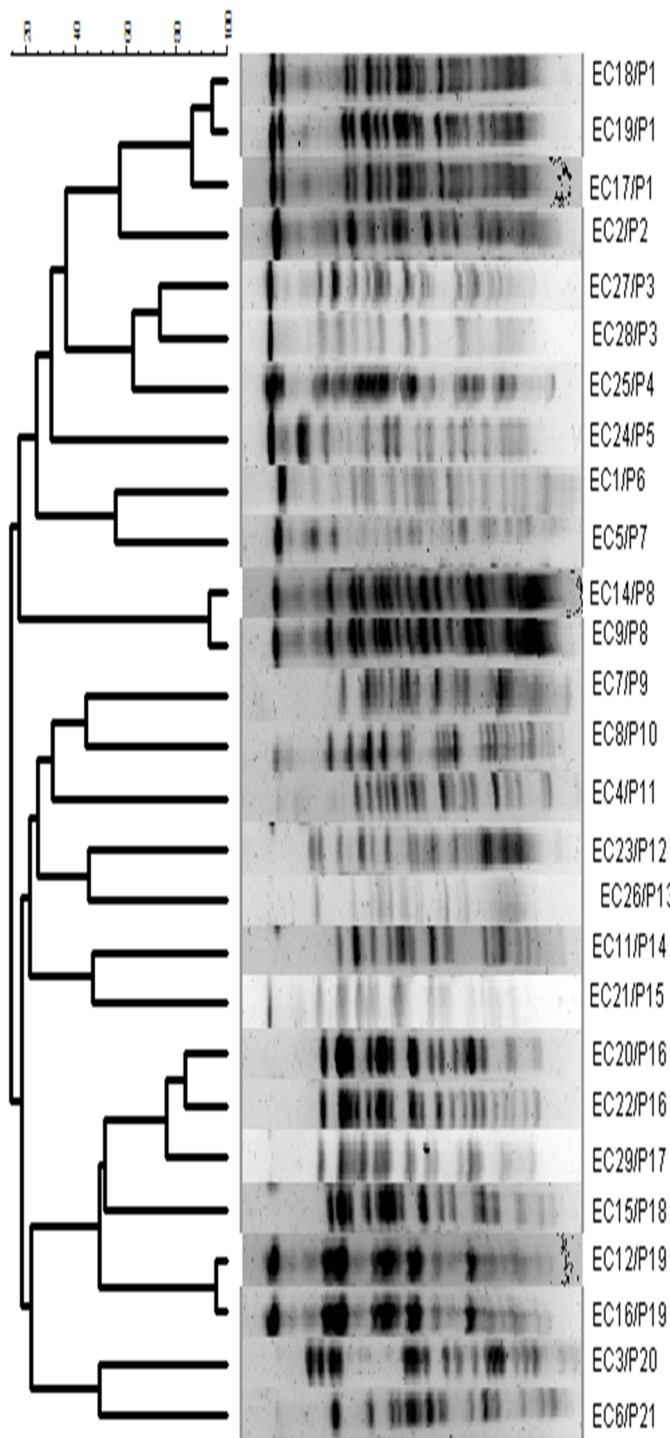

**Figure 2.** PFGE patterns for *E. coli* isolates, with their corresponding pulsotypes.

## 4. Discussion

The aim of our investigation was to determine the ESBL producer-relevant agents and to decrypt their genetic factors. The ubiquity of ESBLs produced by *Enterobacteriaceae*, particularly with *E. coli* (ESBL-EC), has increased worldwide throughout the past two decades [25]. In our investigation, *E. coli* isolates were collected from various samples. Several cefotaxime-resistant strains were detected. A total of 92.59% (25/27) strains were considered as being ESBL producers. However, 3.7% (1/27) of isolates produced AmpC β-lactamases. Indeed, this strain contained both ESBL and AmpC β-lactamase enzymes. This rate of ESBL-producer isolates was similar to those reported previously [7] from Al-Abha

and Riyadh, KSA [26,27]. The prevalence rates of ESBL-EC were 6.5% and 10.3% in 2002 and 2004, sequentially [8]. The preponderance rates were 15.4% and 4.5% for inpatients and outpatients, respectively [28,29]

The multidrug resistance (MDR) was described for all isolates, even for quinolones, aminosides, tetracycline, trimethorprim/sulfamethoxazole, fosfomicin, and chloramphenicol. MDR is frequent in *E. coli*; however, the high rates of resistance found for fosfomycin in this study are uncommon in such strains. Conversely, fosfomycin susceptibility tests are limited, since this agent is occasionally available in most clinical laboratories. Uncomplicated UTI is commonly treated in many countries with fosfomycin, whereas worldwide attention has been oriented to sparing carbapenems in ESBL-producing strains. Fosfomycin in combination with colistin was recommended for treating *Enterobacteriaceae* that were resistant to carbapenems. A recent study from China [29] reported uropathogenic ESBL-producing *E. coli* isolates that are resistant to fosfomycin. Indeed, fosfomycin resistance was encoded by the fosA3 gene carried by a 54.2 Kb transferable plasmid also co-harboring a $bla_{CTX-M}$ gene. The genetic characterization of fosfomycin resistance mechanisms and its possible linkage to ESBL-encoding genes needs further investigation in KSA.

The typical TEM and SHV variants of ESBLs have declined during the past two decades and were interchanged worldwide by the CTX-M group as the predominant ESBL group. Similarly, the ESBLs generated in our strains harbored CTX-M β-lactamases. The $bla_{CTX-M-15}$ (21/27), $bla_{CTX-M-27}$ (5/27), and $bla_{CTX-M-9}$ (1/27) genes were identified. The $bla_{CMY-42}$ gene was recorded in two strains; moreover, one isolate co-accommodated the $bla_{CTX-M-15}$ gene. The concomitant occurrence of ß-lactamase coded by the $bla_{SHV-1}$ or $bla_{TEM-1}$ genes was also common in these isolates, as reported previously [30]. Our findings support other studies from KSA showing the prevalence of a $bla_{CTX-M15}$ type [26,31]. However, the $bla_{CTX-M-9}$ and $bla_{CTX-M-27}$ genes are prevalent in KSA, and low rates were reported [7,32]. It is also important to know the occurrence of the $bla_{CMY-42}$ gene, a CMY-2 variant, which was present in two isolates of our collection. The $bla_{CMY-42}$ was first described in *E. coli* [33], and was then rarely reported and mainly found to be associated with $bla_{NDM-5}$, $bla_{SHV-12}$, $bla_{CTX-M-14}$, or $bla_{CTX-M-15}$, as described in one isolate of our study. Presumably, we describe an early case of $bla_{CMY-42}$ harboring *E. coli* isolates in KSA [34].

The prevalent plasmid replicon group was found in resistant *Enterobacteriaceae* segregated from both animals and humans holding incompatibility (*Inc*) group F (including *FIA*, *FIB*, and *FII* replicons), and A/C, L/M, I1, HI2, and N [35,36]. Different $bla_{CTX-M}$ genes are also combined with specific plasmid replicon types (*IncN*, *I1*, *FII*, and *L/M*) [36]. The $bla_{CTX-M-15}$ was primarily found on plasmids that involve FII and FIA replicons, and, to a lesser extent, to *IncI1*, *IncN*, and *IncA/C*, as well as being on pir-type plasmids. Concordantly, *IncF* (3/27) and *IncFIA* (5/27) plasmids were identified among our collection, and both co-occurred in five isolates, indicating the possible localization of the $bla_{CTX-M-15}$ gene on one of these plasmids. The concomitant transfer of the $bla_{CTX-M-15}$ gene with *IncF* and/or *IncFIA* plasmids for these strains supported this hypothesis. However, further experiments (S1-PFGE hybridization) are needed to assess the exact genetic localization of all *bla* genes. Both integrons belonging to classes 1 and 2, as well as the occurrence of these plasmids, might explain the MDRs of our isolates, as reported previously [31,32]. A significant proportion of our isolates were observed to belong to phylogroups D and B2. This finding concurred with other studies; moreover, most of the ESBL-producing *E. coli* are considered to be the main leading causes of extra-intestinal infections. Furthermore, three CTX-15-producing *E. coli* strains positive for the B2 phylogroup belonged to the ST131 clone. The ST131 clone, known as the major pandemic clone, controlled the worldwide dissemination of the β-lactamase CTX-15 type [35,37]. This clone has been also reported previously in KSA [7].

Genetic relatedness assessed by PFGE revealed 21 pulsotypes, where P1 encompassed three isolates, while pulsotypes P8, P16, and P19 contained two isolates. The clonally related isolates were all of pulsotype B2, and all, except two of pulsotype P16, harbored identical

β-lactamases. However, antimicrobial susceptibility, PMQR genes, integrons, plasmids, and virulence genes contents were not mainly identical within the related isolates.

Taken together, despite the reduced isolate number, our study showed the dominance of the CTX-15 enzyme among *E. coli* in KSA, and the occurrence of some clonal isolates spreading within patients from different regions. ST131 was not the main clone in our isolates; other lineages or plasmids contributed significantly to the spread of the CMY-42 enzyme.

## 5. Conclusions

We describe the first report of CMY-42-producing *E. coli* isolates in the Kingdom. The CMY-42 enzyme was identified among unrelated *E. coli* pulsotypes, as well as CTX-M-27, CTX-15, and CTX-9. The ST131 seems to be a significant contributor for the spread of the CTX-15 enzyme among our isolates. The *IncF* and/or *IncFIA* plasmids were identified within the $bla_{CTX-M-15}$ gene, as well as class 1 and class 2 integrons, showing their probable horizontal transmission. The MDR phenotype associated with several virulence genes was observed in our collection, showing a large dissemination of antibiotic resistance in the eastern region. Our research could form a strong guide for clinicians who are interested in the genetic factors of MDR.

The rigorous control of antibiotic therapy and the continuous surveillance of epidemiologic analysis are essential for limiting the resistance diffusion factor in the post-COVID-19 era, as well as fostering a better understanding of their dissemination process for better oriented outcomes. The actual situation is very difficult due to the increased need for non-pharmaceutical (sanitizers) and pharmaceutical (antibiotics) stressors used during the COVID-19 era. Antibiotic resistance management within artificial intelligence and machine learning may be powerful tools for helping decision makers to implement new solutions in healthcare departments and medication prescriptions.

**Author Contributions:** Conceptualization, M.B.A.; methodology, M.B.A. and S.F.; software, M.B.A. and S.F.; validation, M.B.A. and S.F.; formal analysis, M.B.A. and S.F.; investigation, M.B.A. and N.A.-S.; resources, M.S.A., I.B., and N.A.-S.; data curation, M.B.A.; writing—original draft preparation, M.B.A., S.F., and M.S.A.; writing—review and editing, M.B.A.; visualization, M.B.A.; supervision, M.B.A.; project administration, M.B.A. All authors have read and agreed to the published version of the manuscript.

**Funding:** This research was funded by King Faisal University, grant number GRANT593, and APC was funded by the Deanship of Scientific Research of King Faisal University, project number GRANT593.

**Institutional Review Board Statement:** The study was conducted in accordance with the Declaration of Helsinki and approved by the Ethics Committee of the Deanship of Scientific Research at King Faisal University (EA000528).

**Informed Consent Statement:** Patient consent was waived due to retrospective data.

**Data Availability Statement:** The data presented are available on request from the corresponding author.

**Acknowledgments:** All the authors are grateful to the Deanship of Scientific Research at King Faisal University, project number GRANT593.

**Conflicts of Interest:** The authors declare no conflict of interest.

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
