# Peer review of "Characterization of Escherichia coli Cefotaxime-Resistance in Al-Ahsa, KSA: Predominance of CTX-15 and First Report of blaCMY-42 Gene"

_applsci, doi:10.3390/app12199964_

Round 1

Reviewer 1 Report (New Reviewer)

Article - Characterization of Escherichia coli Cefotaxime-Resistant in Al Ahsa, KSA: Predominance of CTX-15 and First Report of blaCMY-42 Gene

Comments

Page 2- line 56= Reference is missing

Page 2- line 73= the sentence needs a full stop

Page 3- line 82 = topic can be changed to “characterization of antibiotic resistant gene”

Page 3- line 107 = correct the word “urines”

Page 3- line 117 = Explain what ‘both’ are.

Page 4- line 120 = topic can be change to ‘characterization of antibiotic resistance’

Page 4- line 131-134 = Some antibiotics are started with capital letters, some in simple

Page 5- line 138 = the sentence “Plasmid replicon type was identified in 48.14%” is not very clear. Change it

Page 6- figure 2 = Please modify the figure description in a more simplified manner.

Page 8- lines 201-202 = The sentences have to be clearer and easier to understand

Page 9- line 244 – Change the sentence containing ‘showing it’. Change the sentence into a more professional one.

Overall, the writing style has to be improved, the sentences seem to have complexed nature.

Author Response

Reviewer 2 Report (New Reviewer)

Author Response

Reviewer 3 Report (New Reviewer)

The manuscript by Aissa et al provides novel insights about cefotaxime-resistant E. coli strains and the first report of the blacmy-42 gene in the Kingdom of Saudi Arabia. The research seems to be well planned and nicely executed. The results are clearly articulated and I personally think the study will be of interest to a wide readership (specifically for people working on antibiotic resistance).

However, I do have the following suggestions for the authors

1. The abstract introduction and discussion need extensive English editing, I have encountered numerous grammatical errors and at times it's hard to understand what the authors want to articulate. I recommend the authors to get the manuscript edited by a native English speaker. I believe this will help increase the quality of the manuscript substantially. 

2.  Do the authors think they should have included more isolates in the study? I do think the numbers are less. I would like the author's take on this.

Author Response

Reviewer 4 Report (New Reviewer)

Congratulate the authors for their work. The presented manuscript is scientifically significant and impressive. This research will be of high significance and will attract readers interested in this field. I reviewed the manuscript in details and several times. I would highly recommend the publication of this manuscript after a minor revision.

- Add more details about the antimicrobial susceptibility tests and Cefotaxime resistance transfer assays.

Round 2

Reviewer 3 Report (New Reviewer)

After the extensive English editing, I believe that the quality of the manuscript has increased substantially. I think it should be accepted in the present form. 

This manuscript is a resubmission of an earlier submission. The following is a list of the peer review reports and author responses from that submission.

Round 1

Reviewer 1 Report

This manuscript still needs signficant work. The text and tables are hard to follow. "This is a table" definitively should not be the title of a table.

Reviewer 2 Report

In this manuscript, the authors determined the genetic factors and plasmids of ESBL producing Escherichia coli strains, isolated from various samples in KSA. They also described the genetic relationships and virulence factors of isolates. Researchers found that more than 90% of tested strains were ESBL producers. They also evaluated the occurrence of antibiotic resistant isolates, showing the high prevalence of MDR. Moreover, researchers tested the transfer of genes responsible for antibiotic resistance through the process of conjugation. As authors suggesting, this research is the first report of CMY-42-producing E.coli in KSA. In summary, this manuscript underline the rapid spread of ability to ESBL production among bacteria, pointing the need of antibiotic therapy control, especially at the post-COVID era, were antibacterial agents were overused. It was nice to read, however publication needs some improvements:

Major:

Methodology should be described more precisely. For example: Method 2.2. What test of synergy was used? How was the conjugation performed? In phylogenetic analysis used method should be mentioned in methods, not only in the figure description. 

Minor:

Line 32 –it should be COVID-19 – capital letters

Line 33 – it should be SARS-CoV-2

Line 36,49,56,201,204,224,241 – double space

Figure 1 – graphic should be placed more on the left or centered, now it is cut off on the right side. I also recommend to use brighter color for the text background

Figure 2 – the scale above dendrogram is barely visible, I think that this graphic could even take the whole page

Line 161 – I guess that there should be some description of the table

Used names should be unified. Sometimes genes names are written without cursive (e.g. lines 136,137, 155-158). Additionally it would be nice to use the same abbreviations in the whole text and explain it at the first time it is used like “KSA - Kingdom”.

Reviewer 3 Report

The manuscript describes the characterization of cefotaxime-resistant E. coli clinical isolates in Al-Ahsa, KSA. There are relevant and interesting results but the manuscript needs to be strongly improved. The writing is unclear and hard to read, the manuscript has errors, and some data is contradictory. There terms cefotaxime-resistance and ESBL-producer are sometimes wrongly used. References format has several mistakes. English should be improved and accurate terminology should be used. Only some comments are below.

Abstract Section.

-        Lines 16-17. The total number of E. coli isolates should be included.

-        Lines 20-21. Proportions of each blaCTX-M genes should be included, like PMQR genes.

Introduction Section.

-        Lines 32-34. Salmonella and Shigella are not associated with secondary infections among SARS COV-2-patients (References 2 and 3). However, these species can cause infections in international travelers (Reference 4). The sentence should be rewritten.

-        Wrong terminology use: Line 36: “delivering”; Line 43: “unlimited diffusion”; Line 44: “continuous outbreak”; Line 48: “for the remedy”; Line 54: “The ESBL epidemiology is related”.

Materials and Methods Section.

-        2.1 Study Design. Line 71 and Figure 1. The exclusion of EC10 and EC13 is not clearly explained. They are susceptible to C3G (line 71, higher sensitivity?) but they are ESBL-producer (Figure 1)?. If it is the case (susceptible to C3G and ESBL-producer) they should be included and characterized, because if they are ESBL-producers then should be considered as resistant to C3G.

-        Figure 1. It has no sense the place where are located the dates: November 2016 and March 2017. What means Double check?

-        Line 77-78. “The strain origins were saved.” It is not clear.

-        Lines 92-94. Terms “used within the”, “clonal correlation” are wrong.

Results Section.

-        3.1. Lines 104-106. The sentence should be rewritten.

-        3.1. Lines 106-107. The sentence: “The multidrug resistance …”. It is confusing, the collection contains non-ESBL producing isolates?

-        3.2. Lines 119-120. How many CTX-M beta-lactamases were characterized? 27 or 28?. At the discussion section, lines 174-175, the authors describes 25 ESBL producers.

-        3.2. Lines 122-123. CMY-42, TEM-1 and SHV-1 enzymes are not ESBL!

-        Table 2. Headline: “non-ESBLs E. coli strains”, is ok?

-        Lines 120-127. blaTEM-1 is widely disseminated, therefore looking for association with this enzyme is incorrect.

Discussion Section. Several sentences are confusing, erroneous written and some results are wrongly interpreted.

-        Lines 174-176. These sentences are not clear. For example, a strain harbors both and ESBL plus AmpC, but it was included among the 25 isolates or not?

-        Lines 176-177. Which is the ESBL-producer rate?

-        Lines 183-184. Sentence: “MDR is frequent…” is unclear.

-        Line 196. Sentence: “Similarly, …”is unclear.